# Towards the Detection of GPS Spoofing Attacks against Drones by Analyzing Camera’s Video Stream

**DOI:** 10.3390/s22072608

**Published:** 2022-03-29

**Authors:** Barak Davidovich, Ben Nassi, Yuval Elovici

**Affiliations:** Department of Software and Information Systems Engineering, Ben-Gurion University of the Negev, Beer-Sheva 8410501, Israel; nassib@post.bgu.ac.il (B.N.); elovici@bgu.ac.il (Y.E.)

**Keywords:** drones, GPS spoofing, countermeasures

## Abstract

A Global Positioning System (GPS) spoofing attack can be launched against any commercial GPS sensor in order to interfere with its navigation capabilities. These sensors are installed in a variety of devices and vehicles (e.g., cars, planes, cell phones, ships, UAVs, and more). In this study, we focus on micro UAVs (drones) for several reasons: (1) they are small and inexpensive, (2) they rely on a built-in camera, (3) they use GPS sensors, and (4) it is difficult to add external components to micro UAVs. We propose an innovative method, based on the video stream captured by a drone’s camera, for the real-time detection of GPS spoofing attacks targeting drones. The proposed method collects frames from the video stream and their location (GPS coordinates); by calculating the correlation between each frame, our method can detect GPS spoofing attacks on drones. We first analyze the performance of the suggested method in a controlled environment by conducting experiments on a flight simulator that we developed. Then, we analyze its performance in the real world using a DJI drone. Our method can provide different levels of security against GPS spoofing attacks, depending on the detection interval required; for example, it can provide a high level of security to a drone flying at altitudes of 50–100 m over an urban area at an average speed of 4 km/h in conditions of low ambient light; in this scenario, the proposed method can provide a level of security that detects any GPS spoofing attack in which the spoofed location is a distance of 1–4 m (an average of 2.5 m) from the real location.

## 1. Introduction

A Global Positioning System (GPS) is a satellite-based radio navigation system that provides geolocation and time information. GPS sensors are located in various devices and vehicles such as cars, planes, cell phones, ships, large and small UAVs, and more. In light of this, we propose a novel GPS spoofing attack detection system. We focus our solution on small UAVs (drones) for several reasons: (1) they are small and inexpensive, (2) they rely on a built-in camera, (3) they use GPS sensors, and (4) it is difficult to add external components to micro UAVs. Many attacks on GPS sensors attempt to spoof the location of the device at a given moment, including GPS spoofing, which is one of the most common attacks of this kind. A successful spoofing attack enables the attacker to perform various attacks, such as changing the direction of the device, redirecting the device to another location in the world, and more. A drone manufacturer can detect GPS spoofing attacks by analyzing the physical layer of the signal, because it has access to this data via the sensor. Currently, subtle GPS spoofing attacks are not detected by drone manufacturers, creating a real problem for consumers who need an effective user-level solution (i.e., a solution that is effective given the existing constraints whereby the consumer likely has limited to no access to the relevant data). Recent studies have suggested countermeasures to detect, mitigate, and prevent GPS spoofing attacks, however the proposed methods have numerous disadvantages, and as a result, a recent SoK identified GPS spoofing attacks against drones as a scientific gap [1] that threatens drones’ ability to perform their tasks.

In this research, we examine whether a drone’s video stream can be used to detect GPS spoofing attacks, without the need for additional drone hardware or memory, or prior knowledge of the flight area. We propose a method capable of detecting GPS spoofing attacks by verifying the measurements obtained by the GPS sensor against the video stream captured by the drone’s camera. We analyze the suggested method’s performance in a controlled environment by conducting experiments on a flight simulator that we developed. Then, we analyze its performance in the real world using a commercial drone. We show that our method can detect any attempt to launch a GPS spoofing attack in which the spoofed location is a distance of 1–4 m (an average of 2.5 m) from the real location and for a drone flying at altitudes of 50–100 m over an urban area at an average speed of 4 km/h with different levels of ambient light.

The significance of our method with respect to the methods proposed in related work is as follows:Our method relies on existing hardware: in contrast to methods presented in other studies (e.g., [2,3,4]), our method does not involve the use of additional hardware, which also makes it cost-effective;Our method is database independent: in contrast to methods presented in other studies (e.g., [5]) which use a precompiled database, our method does not rely on a precompiled database or a map of the drone’s flight area;Our method offers flexibility: unlike other methods, it can be implemented on the drone itself or from the ground control station used to control the drone (i.e., on the drone’s controller);We empirically evaluate the accuracy of our method and determine the level of security for a situation in which the spoofed location is an average of 2.5 m away from the actual location, an aspect that was not evaluated in related studies.

In summary, this paper makes the following contributions: (1) We propose a new method for the detection of GPS spoofing attacks on devices that include (or contain) an on-board video camera. The method’s performance was evaluated with a simulator and in a real life scenario. (2) We develop a simulator that allows us to simulate a drone hovering over a Google Earth map, allowing us to simulate flights in different conditions (various altitudes, terrains, speeds, and level of ambient light) and locations. The code of the simulator has been uploaded to GitHub [6].

The remainder of this article is organized as follows. We provide an overview of related work in Section 2. The proposed method is described in Section 3. In Section 4, we discuss the analysis performed in our simulation environment. The results of our real-world evaluation are presented in Section 5. In Section 6, we present the method’s limitations. In Section 7, we discuss our plans for future work. Our article concludes in Section 8, where we summarize our findings.

## 2. Related Work

In this section, we review prior studies that proposed countermeasures against GPS spoofing. The GPS protocol is vulnerable to spoofing attacks, since it lacks encryption and authentication mechanisms. As a result, attackers can inject false GPS signals using software-defined radio (SDR) [7] or dedicated GPS spoofers (which can be purchased online), causing the drone to believe that it is flying in a location that differs from the actual location. Studies have shown that GPS spoofing attacks against drones can cause a drone flying in autonomous mode to accelerate in the attacker’s chosen direction [8] (by transmitting fake GPS coordinates in the opposite direction); force a drone flying in manual mode to land [9] (by sending a no-flight zone alert that triggers a safety mechanism, causing the drone to land); or change the direction of the drone’s movement itself [10].

Other studies have suggested countermeasures against GPS spoofing attacks by integrating additional hardware. One study [3] suggested the use of a multi-receiver for GPS spoofing detection to detect malicious fake signals, by verifying the GPS measurements using the fixed distances between the receivers and then measuring the distances between the receivers’ reported locations. When the GPS signal is legitimate, the distance will be similar to the fixed distances, but when there is a GPS spoofing attack, the measured distances will be very close to zero, as all the receivers are spoofed with the same fake location; this method would be difficult to implement with small drones, because additional hardware is needed for all of the GPS receivers. SPREE [4], a method presented in another study, is a countermeasure for GPS spoofing attacks that can also detect takeover attacks; it relies on the auxiliary peak method, which is used in combination with a navigation message inspector in which the strongest satellite signal as well as other weaker environment signals are tracked. SPREE’s main disadvantage is that external hardware is needed. Another study [11] suggested the use of Iridium signals to detect Global Navigation Satellite System (GNSS) spoofing. The authors reverse engineered parameters from the Iridium satellite constellation, such as the satellite’s speed, packet interarrival times, maximum satellite coverage, satellite pass duration, and satellite beam constellation. With those parameters, they proposed a solution for the detection of a target user’s deviations from his/her path caused by a GNSS spoofing attack. The main disadvantage of the proposed method is the requirement to provide the parameter values for each user, which may be a barrier to a consumer without this capability. Another study proposed SemperFi [12], a single antenna GPS receiver capable of tracking legitimate GPS satellite signals and estimating the drone’s true location (even during a spoofing attack); this method relies on a comparison of satellite signals’ time of arrival (ToA) in order to determine whether a GPS spoofing attack occurred. This solution’s main disadvantage is that the consumer is dependent on the manufacturer, which is responsible for the solution’s implementation, and cannot implement the solution him/herself. In a paper presenting a similar solution that deals with the GPS signal [13], the authors proposed a receiver that uses maximum plausibility estimates after the validator signal is dropped, in order to assess the correct location.

Other studies suggested countermeasures that do not rely on GPS for navigation and instead rely on the cellular network; for example in [14], the authors presented “drive me not”, a GPS spoofing detection method that utilizes mobile cellular network infrastructure to validate the position received by the GPS infrastructure. Other research [15] utilized UpLink’s received signal strength (RSS) measurements for cross-position validation for GPS spoofing detection. In another study [16], the authors built a network of clustered ground base stations (BSs) that cooperatively serve a number of UAV-UEs. The main disadvantage of these solutions is that they will only work with drones that have cellular communication capabilities; in contrast, our solution relies on a camera, which is a more common drone component.

Methods for navigating without GPS have also been proposed, such as MVP [17], a method for navigating in locations such as tunnels and underpasses. The main idea is to extract magnetic fingerprints from geomagnetic field anomalies and compare the measurements against a magnetic map. This method will be difficult to apply in open places and without prior knowledge of the area.

Other research suggested countermeasures that use existing hardware. Several studies proposed methods that use motion sensors and compasses to detect GPS spoofing attacks. For example, one study [18] presented a method that uses gyroscope measurements to verify GPS measurements; its drawbacks are that the sensor’s measurements suffer from false negative/positive errors and must be calibrated in advance. Another method [19] uses an on-board camera and the inertial measurement unit (IMU) to obtain the velocity and position of the drone to detect unexpected changes in the flight path. In this case, the drawbacks are the same as the previous method; additionally, this method relies on two sensors: the camera and IMU. Another study [20] proposed two methods which must be used together for GPS spoofing detection. Here, the difference in acceleration between the GPS receiver and the accelerometer is used to detect GPS spoofing. This solution’s disadvantage is that the accelerometer can be affected by external factors, and flying or driving conditions can affect GPS spoofing detection performance. A deep learning-based solution which uses images from satellites and compares them to images from a drone’s camera to determine whether the locations match was proposed by [5]; its disadvantage is that it requires initial preparation in the flight area. Moreover, every GPS point in the area must be covered, a requirement which increases the size of the precompiled database significantly. Another study presented DIAT [2], which verifies the data/signals received from a drone’s sensors with data from other nearby drones in order to detect compromised measurements. Its drawback is that it needs to support and protect drone-to-drone communication, which requires development of the communication protocol.

While various countermeasures against GPS spoofing attacks have been suggested, they all have disadvantages. A recent SoK paper [1] identified GPS spoofing attacks against civilian drones as a scientific gap that cannot be prevented by any existing mechanisms with a high technological readiness level [21].

## 3. Proposed Method

In this section, we describe the proposed method for the detection of GPS spoofing attacks on drones. Our method relies on data obtained from two sources: a drone’s video stream and GPS measurements. It is based on the assumption that unlike GPS signals, the drone’s video stream cannot be spoofed. To detect GPS spoofing attacks, our method correlates a drone’s movement, calculated from the GPS signals, with the real-time video stream frames. Based on this correlation and a predefined correlation threshold, our method determines whether a GPS spoofing attack has occurred. The correlation between frames is based on brute-force matcher (BFMatcher, which is used to match the features of the first frame with another frame) [22], with the speeded-up robust features (SURF) [23] feature detector. The algorithm receives two frames and calculates their correlation. The output ranges from 0 to 100, with 100 assigned when the same frames are being compared and zero assigned when completely different frames are compared. The correlation decreases as the similarity between the images decreases. There are several other methods that can be used to calculate the correlation; for example, the authors of [24] compared the use of SURF, scale-invariant feature transform (SIFT), binary robust invariant scalable keypoints (BRISK), and oriented FAST and rotated BRIEF (ORB), and found that SURF was the fastest algorithm and provided good results. For this reason we choose to use BFMatcher with SURF in all of our experiments.

The difference in the correlation between the first frame ($frame_{i}$) and the next *n* consecutive frames ($frame_{i+1}$, …, $frame_{i+n}$) is continuously calculated. Based on the GPS measurements and the similarity correlation between the frames, a model, which is used to verify the location for the next *q* consecutive frames ($frame_{i+n+1}$, …, $frame_{i+n+q}$), is created. For each of the next *q* consecutive frames ($frame_{i+n+1}$, …, $frame_{i+n+q}$), the distance is predicted based on the frame’s similarity correlation with the first frame. If the error, meaning the difference between the actual distance and the distance predicted by the model, is beyond a threshold, the model issues an alert that the GPS measurements do not correlate with the video stream (i.e., a GPS spoofing attack has taken place). In each flight, our method is applied for a specified period of time (from $frame_{i}$ to $frame_{i+n+q}$). The similarity correlation between consecutive frames decreases as a function of the distance, so eventually there will be no correlation between the first and last frames, and a new model will be generated.

Our method includes the following steps:After the drone takes off, frames along with their specific GPS locations are collected. For each second of flight, one frame and its GPS location are saved. Thus, after *n* seconds, we will have *n* frames ($frame_{i}$ = the first frame in the time window, and $frame_{i+n}$ = the *n*th frame in the time window);The similarity correlation between $frame_{i}$ and all other *i* + *n* frames is calculated. At the same time, the distance (based on the GPS measurements) between the location of $frame_{i}$ and the location of all other *i* + *n* frames is calculated. This information is used to generate a graph presenting the correlation vs. distance (an example of such a graph is presented in Figure 1). This graph allows us to model the change in frame correlation against the GPS distance, which provides a suitable function for the prediction of the next point (f(correlation)=distance) on the graph;The next $frame_{i+n+1}$ and its GPS location are obtained, and the correlation between $frame_{i}$ and $frame_{i+n+1}$ (correlation_i_i+n+1) and the distance between the GPS locations are calculated. Then, the model can predict predict_distance from function f(correlation_i_i+n+1) = predict_distance. We then have the actual and predicted distance between $frame_{i}$ and $frame_{i+n+1}$;If predicted_distance≈real_distance, the GPS location correlates with the frame, indicating that there was no GPS spoofing attack on the drone. If predict_distance >real_distance, there is no correlation between the GPS location and the frame, confirming that there has been a GPS spoofing attack.

This process is presented in Figure 2 with an example in which we collected five frames ($frame_{0}$ to $frame_{4}$), along with their GPS locations, and tried to predict the distance for $frame_{5}$ (*i* = 0, *n* = 4).

## 4. Analysis and Simulation

In this section, we describe the experiments and analysis performed in our simulation environment, a setting which allows us to investigate the effect of (1) the drone’s altitude, (2) the drone’s speed, (3) the terrain over which the drone flies, and (4) ambient light on our method’s performance. The simulation environment was designed so that we could examine our solution in a controlled environment, without external disturbances, and demonstrate a proof of concept of our solution in various field conditions. In this setting we can also test our proposed method anywhere in the world by using Google Earth. To predict the next position, we graphed the correlation vs. distance, built a function suitable for the graph, and learned how it behaves. A test window is defined as the number of points needed to construct a linear regression function so that f(correlation)=distance_prediction.

We used the simulator to conduct various experiments aimed at assessing the influence of the specific factors (1–4 in the previous paragraph) on the method’s performance. The following metrics were used to evaluate the performance: the root-mean-square error (RMSE), $R^{2}$, and mean absolute error (MAE).

### 4.1. The Simulator

To build the simulator, we needed to understand how to manipulate the cropped images so they serve as a snapshot from a hovering drone in real time. The cropped size is influenced by the latitude (when altitude increases, more ground is covered) and the camera’s field of view (FOV). We set the camera’s FOV at 79${}^{\circ }$. After a map from Google Earth has been selected, the values of the following parameters are used in the simulation: (1) *metersInLegend*—the scale in meters from the Google Earth legend; (2) *legendPixels*—represents the number of actual pixels, which appears in the Google Earth legend; (3) *photoWidth*—the photo’s width in pixels; and (4) *photoHeight*—the photo’s height in pixels. We uploaded the code to GitHub [6]. Figure 3 presents an example of the simulator screen.

The following steps must be performed when using the simulator: (1) Obtain a map from Google Earth [25], and (2) insert the drone speed and altitude, and click on the “Submit” button. (3) Click on the image; the position clicked on represents the drone’s starting position, and the black frame captures what the drone sees in a specific frame. (4) Use the arrows to move the black frame on the map—each movement will crop an image; the difference in the distance (in meters) between each frame is calculated, and the correlation between each frame is calculated; click the “Enter” button to stop.

We then plot the correlation vs. distance on a graph. The graph starts at the (0, 100) point, which means that at the starting location the distance is zero. When moving along the map, the correlation between the images decreases as a function of the distance.

### 4.2. Influence of Altitude

The altitude affects the efficacy of the test window and how quickly it needs to be changed. As the drone’s altitude increases, its perspective widens, and the changes between two consecutive frames are more significant.

Experimental Setup: In this experiment, we chose three locations in a neighborhood of Manhattan (Figure 4, Figure 5 and Figure 6). In the simulations, the drone flew at three different altitudes (50, 100, and 200 m) at a consistent speed of 5 m/s (meters per second). For each altitude, the simulation ended immediately after the drone was out of the scope of the first frame (this was done in each of the experiments described below), because we found that a correlation of zero is obtained when the drone is out of the first image’s scope due to the fact that there is no longer any similarity to the first image.

Results and Conclusions: The results are presented in Table 1, Table 2 and Table 3. Based on the results obtained, we calculated the RMSE, $R^{2}$, and MAE. Several interesting insights can be derived from the results: (1) RMSE: as the altitude increases, the RMSE value decreases; (2) $R^{2}$: while this tends to be a value of one, it can be seen that the $R^{2}$ value decreases at $10^{-3}$; and (3) MEA: when the value is close to zero we get a better match to reality.

### 4.3. Influence of Speed

The drone’s speed affects the efficacy of the test window and how quickly it needs to be changed. As the drone’s speed increases, its perspective widens, and the correlation between consecutive frames decreases accordingly. To isolate the problem, a constant velocity is used in this experiment.

Experimental Setup: In this experiment, we used three different altitudes (50, 100, and 200 m) and one location in Manhattan (Figure 4). For each altitude, three different speeds and a constant velocity were used. One frame was collected per second. As mentioned in Section 4.2, the simulation ended immediately after the drone was out of the scope of the first image.

Results and Conclusions: Table 4, Table 5 and Table 6 present the results for altitudes of 50, 100, and 200 m. Several interesting insights can be derived from the results: (1) RMSE: as the speed increases, the RMSE value decreases; (2) $R^{2}$: while this tends to be a value of one, it can be seen that the $R^{2}$ value decreases at $10^{-3}$; (3) MEA: when the value is close to zero we get a better match to reality; and (4) by examining each speed separately, at higher altitudes the error associated with high speeds decreases. We observe that the best speed for this terrain is 10 m/s, where it can be seen that the values of the RMSE, $R^{2}$, and MEA are the lowest for all altitudes.

### 4.4. Influence of Terrain

When the terrain over which the drone flies changes constantly (as it does in urban areas), the correlation between the frames has greater influence; in contrast, in unsettled open/flat areas there are limited changes in the correlation between consecutive frames. The results presented in Section 4.2 demonstrate that the proposed method provides good results in urban environments, so in this section, we examine only unsettled open/flat areas.

Experimental Setup: In this experiment, the drone flew over an unsettled open/flat area at the lake in Central Park (Figure 7). Using a histogram normalized for the color scale, the flatter the histogram, the more stable the image changes are. As the starting point, we used the worst-case scenario in which the drone hovers over the lake at three different altitudes (50, 100, and 200 m; see Figure 8) at speeds of 5, 10, and 20 m/s. As mentioned in Section 4.2, the simulation ended immediately after the drone was out of the scope of the first image.

Results and Conclusions: Table 7, Table 8 and Table 9 present the results for this terrain for altitudes of 50, 100, and 200 m. Based on the results obtained, we calculated the RMSE, $R^{2}$, and MEA. The results obtained for the three metrics indicate that none of the metrics were suitable for evaluating the influence of the terrain on our method’s performance. However, several interesting insights can be derived from the results: (1) RMSE: the RMSE value is very high (over 180) at all speeds and altitudes; (2) $R^{2}$: the value obtained for this metric is never close to one; therefore the error keeps increasing; and (3) MEA: the MEA value is never close to zero. Based on the results of this experiment, we conclude that our solution is not effective when applied on unsettled open/flat terrains; it is more suitable for urban terrain.

### 4.5. Influence of Ambient Light

Ambient light can influence how the terrain trajectory changes. We note that our method relies on the video stream from a drone’s built-in camera, which does not provide any special night vision capabilities or the ability to see in total darkness.

Experimental Setup: To obtain images with different levels of daylight, which are not available from Google Earth, we used an image processing technique commonly used to darken images [26]. This allowed us to simulate four lighting conditions: 75%, 50%, 25%, and 10% light (see Figure 9) at altitudes of 50–200 m. In this experiment, one location was used (location #1), along with the four lighting conditions. As mentioned in Section 4.2, the simulation ended immediately after the drone was out of the scope of the first image.

Results and Conclusions: Once again, several interesting observations can be derived from the results, which are presented in Table 10, Table 11 and Table 12: (1) RMSE: we observe that when there is less light (reflected in the darkened images), the RMSE value increases; (2) $R^{2}$: as the amount of light decreases, we observe a slight decrease in the values for this metric; (3) MEA: as the amount of light decreases, the MEA value increases; (4) for all lighting conditions, the best results obtained are for an altitude of 200 m; (5) poorer results are obtained when the lighting level is under 25%. Our solution relies on the fact that there is a change in terrain for every frame transition; these changes cannot be detected in dark conditions, which is why the ambient light affects our solution.

## 5. Real-World Evaluation

In this section, we describe our real-world evaluation of the proposed method using a DJI Mavic 2 Pro drone. Due to local regulations, we had to limit the drone’s flight altitude to 100 m; therefore the flights were performed at altitudes of 50 and 100 m. The first step was to capture the GPS locations and their corresponding video frames from the drone’s video stream. To do so, we built an app that collects this information using DJI’s Android mobile software development kit (SDK) [27]. The ability to downsample or oversample the video stream allowed us to simulate flight speeds that differ from the actual speed of the drone. We gathered the samples, which will serve as input to our model, and stored them on a PC.

### 5.1. The Experiment

We performed drone flights in two urban areas (see the flight routes for locations #1 and #2 from an altitude of 50 m in Figure 10 and 100 m in Figure 11). We performed two types of experiments; in one experiment the route was in the shape of a star (simulating a pizza delivery drone that returns to its base station after each delivery to collect a hot pizza for delivery), and in the other experiment a different, non-star-shaped route (simulating a case in which a few packages are being delivered by the drone, as might be done with an Amazon delivery drone) was used. We obtained a total of 5496 frames from an altitude of 50 m and 5425 frames from an altitude of 100 m, each of which was associated with a specific GPS location at a point in time. For simplicity, the drone flew at a speed of approximately 4 km/h, and one frame and its GPS location were obtained per second.

A test window is defined as the number of points needed to construct a linear regression function so that f(correlation)=distance_prediction. The subsequent frames can be predicted by correlating frames with the distance for the same window. In each experiment, we used different window sizes to create the linear regression and calculated the predicted difference in the GPS location distance. We focused on three future points and estimated their position based on the linear regression function, as doing so would allow us to raise an alert within a reasonable amount of time in the case of a GPS spoofing attack. Every calculation window utilizes resources, so we aimed to define a calculation window that optimizes the number of calculations and the false positive rate (FPR), which we would like to keep low.

### 5.2. Results

We present the results of our experiments for a drone flying at altitudes of 50 and 100 m. First, we calculate the average prediction error and maximum prediction error for each time window. Then, for each flight we plot the mean of the average prediction errors and maximum prediction errors at both altitudes (see Figure 12 and Figure 13).

We observe that when the time window is small, the average prediction error is high. For example, in Figure 12 the window size is two (meaning two frames are used to build the linear regression function); in this case, the average prediction error is less than one, but the maximum error is high. In addition, in the figures we can see that in the middle frames the maximum error is at the lowest level, and it remains constant for several window sizes; when the window size grows, the error also starts to increase.

Next, we examine the FPR for specific window sizes in which the maximum prediction error remains constant. For an altitude of 50 m, the maximum error distance is 6 m for all window sizes, and for an altitude of 100 m, the maximum error distance is 5 m for all window sizes. The best results for an altitude of 50 m are for window sizes 4 and 5, where we obtain the lowest error rate for a distance of 4 m (Figure 14). While for 100 m, the best results are for window sizes 5 and 6, where we obtain the lowest error rate for a distance of 3.5 m (Figure 15). False positive alarms may be the result of: (1) the drone’s rotation, or (2) an error in the camera’s autofocus function, which can result in a blurry frame. To demonstrate this, let us examine two examples from real drone flights. In Figure 16, we can see that the drone is rotating. As a result of this rotation, the frame’s similarity decreases to 28, and therefore the correlation between the frames decreases. This causes the model to predict that the distance between the frames (the left and the middle images in Figure 16) is 15 m instead of 1.5 m, which leads to a false positive. In Figure 17, we can see that the middle frame is blurry, and the rest of the frames are clear; as a result of the frame’s blurriness, its similarity decreases to 26, and therefore the correlation between the frames decreases. Again, this causes the model to predict that the distance is 9 m instead of 1 m, which leads to a false positive.

We also investigate how we can determine the optimal window size for our method, the window size that will minimize both the maximum and average errors. For that, we can use the equation: $e\left(\alpha \right)=\alpha *average_{error}+\left(1-\alpha \right)*maximum_{error}$. In Figure 12 and Figure 13, we can see that for an altitude of 50 m the maximum and average errors are obtained for window sizes 4 and 5, while for 100 m they are obtained for window sizes 5 and 6.

Thus, for the following configuration, the proposed method can provide a high level of security, detecting any GPS spoofing attack in which the spoofed location is a distance of 1–4 m (an average of 2.5 m) from the real location. Given this, we conclude that the proposed method is capable of protecting a delivery drone from GPS spoofing attacks and could therefore be used for this purpose.

## 6. Limitations

Our method has a few limitations. In Section 4.4, we saw that the terrain influences our method’s performance. If the terrain is relatively unchanging (e.g., in the case of an ocean or desert), the method’s performance is not optimal. In such terrain conditions, the changes between the frames are almost imperceptible, and as a result, the method will be unable to differentiate between frames. It should be noted that for the use case presented in this article, where we are concerned with the protection of a delivery drone in an urban area, this does not pose a limitation, because for the route of a drone traveling over urban terrain, there is variation between frames.

Another limitation is associated with the ambient light. In Section 4.5, we saw that when the light decreases, the efficacy of our method decreases, to the point that it is not effective in conditions of total darkness. We hypothesize that using a camera with night vision capabilities will increase the method’s efficacy in such conditions.

Another limitation is the altitude. In Section 4.2, we saw that when the altitude is low, the efficacy of our method decreases. Drone altitudes under 50 m can affect the efficacy of our method, since the camera’s FOV will be narrowed to a view that makes it difficult to use the frames. We hypothesize that using a camera with a wide FOV will increase the method’s efficacy.

## 7. Future Work

Extreme weather conditions (e.g., snow, rain, fog) may affect the accuracy of the proposed method, since such conditions can affect the quality of the images captured by the video camera. Additional research could be performed to validate the robustness of our method in extreme weather conditions.

In addition, other types of video cameras may improve the accuracy of the proposed method. For example, a thermal camera may provide improved accuracy in extremely dark conditions. In future research, the method’s robustness with frames obtained from other kinds of cameras could be examined.

We also note that in our real-world tests the drone’s speed was slower than the speed used in the simulator. Our ability to fly the drone at a high speed was limited, as we were instructed by our institution to fly the drone in a controlled manner, so as not to pose a risk to the environment. In future work, we suggest evaluating the method’s robustness at higher speeds in a real-would experimental setting.

## 8. Summary

GPS spoofing attacks can cause dangerous navigation problems. Currently, product manufacturers only offer chip-level solutions, which do not provide a satisfactory means of detecting or preventing such attacks. In this article, we focused on devices that contain video cameras and GPS sensors and proposed a real-time method for the detection of GPS spoofing attacks on drones, which can be used by consumers. The proposed method makes use of the video stream captured by a drone’s camera and the measurements obtained by the GPS sensor. We demonstrated the proposed method on delivery drones, since their role in everyday life is expanding; given the ease with which GPS spoofing attacks can be performed, amateurs may try to attack delivery drones in order to steal the goods they transport. Our evaluation results demonstrate our method’s ability to protect delivery drones from GPS spoofing attacks. Our findings show that it can provide a high level of security to a drone flying at altitudes of 50–100 m over an urban area at an average speed of 4 km/h in conditions of low ambient light; in this scenario, the proposed method can provide a level of security that detects any GPS spoofing attack in which the spoofed location is a distance of 1–4 m (an average of 2.5 m) from the real location. Our method’s advantages include the fact that it does not require any extra hardware or prior knowledge on the flight area.

## Figures and Tables

**Figure 1 sensors-22-02608-f001:**
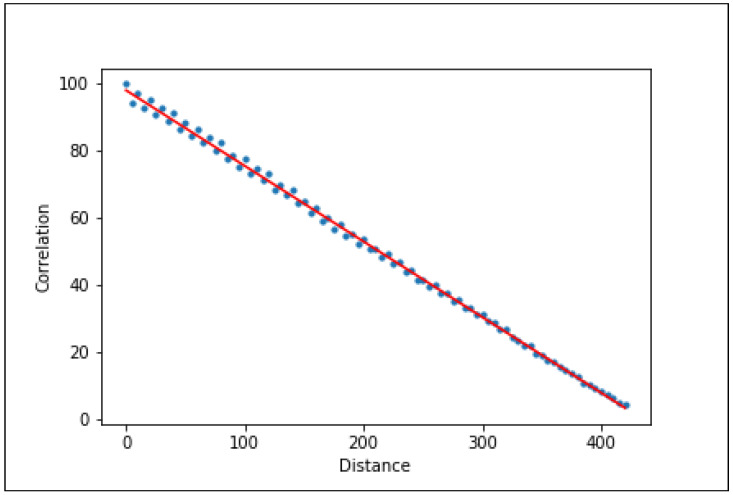
Correlation as a function of distance for location #1 (a neighborhood in Manhattan) for an altitude of 200 m (the blue points indicate the simulation results, and the red line represents the linear regression).

**Figure 2 sensors-22-02608-f002:**
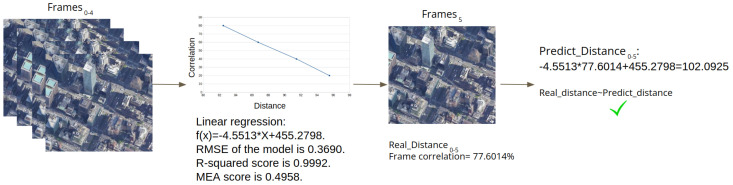
The implementation of the proposed method, using an example in which we obtained five frames ($frame_{0}$ to $frame_{4}$) and tried to predict the distance for $frame_{5}$ (*i* = 0, *n* = 4).

**Figure 3 sensors-22-02608-f003:**
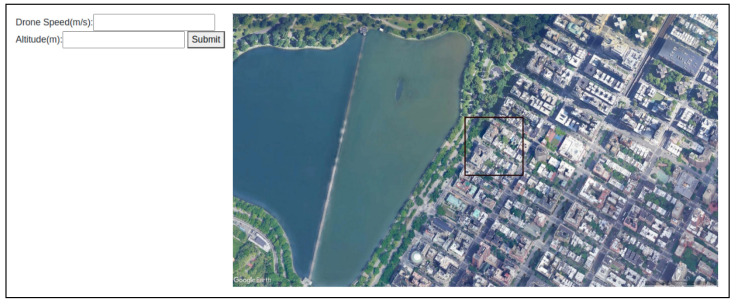
Example of the simulator screen. The drone speed and altitude values are inserted on the left and submitted by the user (by clicking on the “Submit” button); the user clicks on the image and uses the arrows to move the black frame on the map; to finish, the user presses “Enter”.

**Figure 4 sensors-22-02608-f004:**
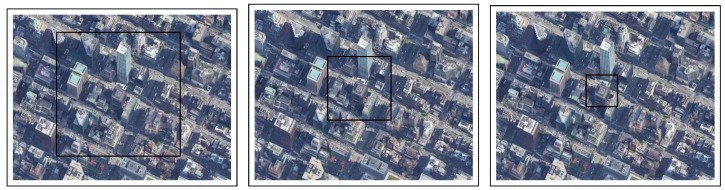
Location #1: 200 m (in the **left** image), 100 m (in the **center** image), 50 m (in the **right** image).

**Figure 5 sensors-22-02608-f005:**
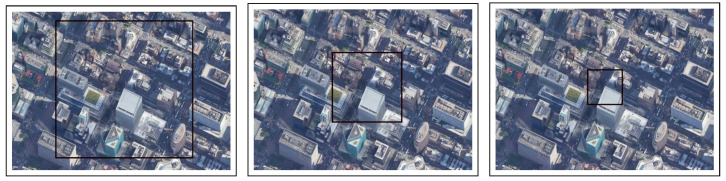
Location #2: 200 m (in the **left** image), 100 m (in the **center** image), 50 m (in the **right** image).

**Figure 6 sensors-22-02608-f006:**
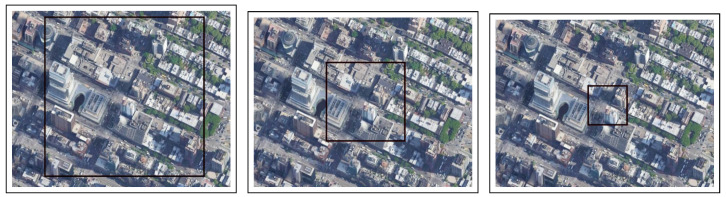
Location #3: 200 m (in the **left** image), 100 m (in the **center** image), 50 m (in the **right** image).

**Figure 7 sensors-22-02608-f007:**
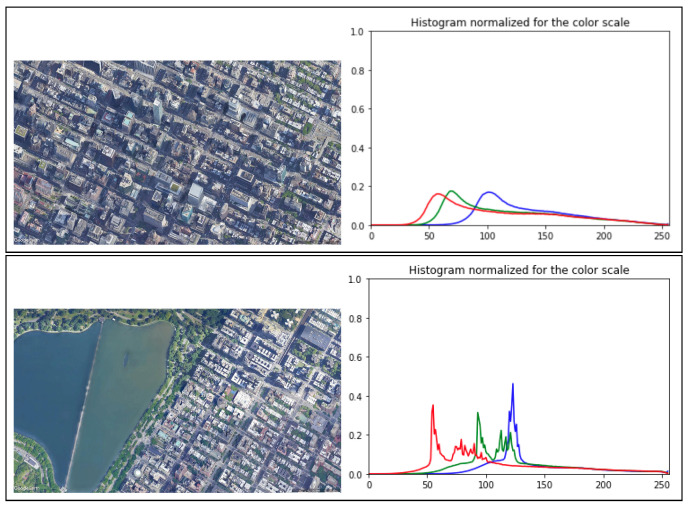
The two types of terrain from an altitude of 1.2 km and histogram graphs: Manhattan as urban (**top**) and Central Park as unsettled open/flat area (**bottom**).

**Figure 8 sensors-22-02608-f008:**

Terrain test for the Central Park Lake location (location #2) from altitudes of: 200 m (in the **left** image), 100 m (in the **center** image), 50 m (in the **right** image).

**Figure 9 sensors-22-02608-f009:**
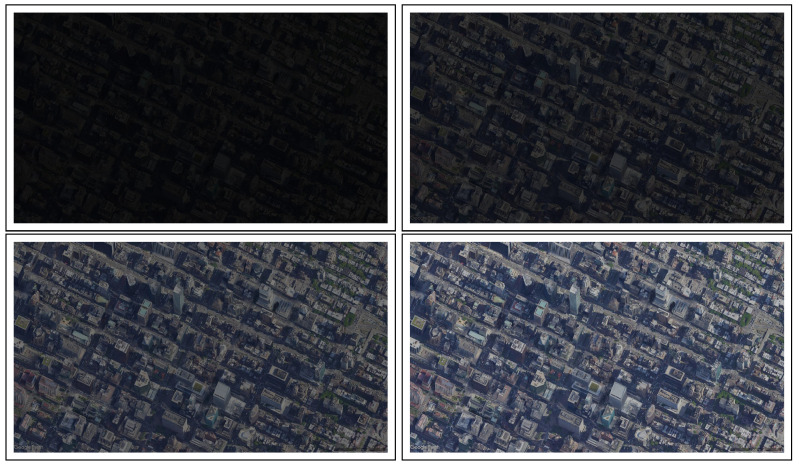
The view of an urban location (location #1) from an altitude of 1.2 km in various lighting conditions (ranging from 75 to 10% light).

**Figure 10 sensors-22-02608-f010:**
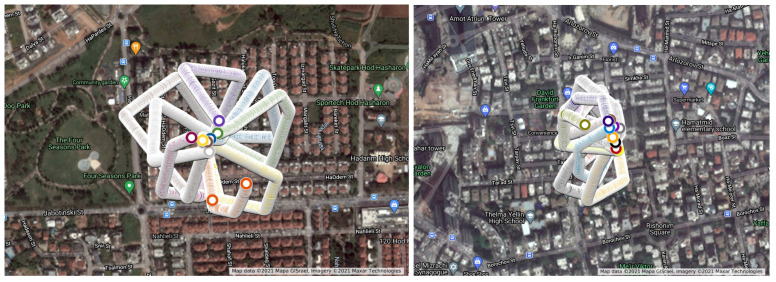
The drone flight route for location #1 (the **left** image) and location #2 (the **right** image) from an altitude of 50 m.

**Figure 11 sensors-22-02608-f011:**
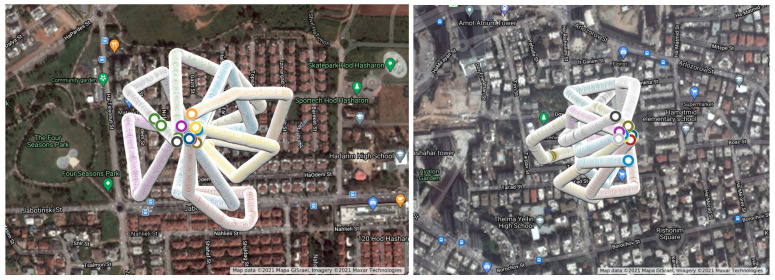
The drone flight route for location #1 (the **left** image) and location #2 (the **right** image) from an altitude of 100 m.

**Figure 12 sensors-22-02608-f012:**
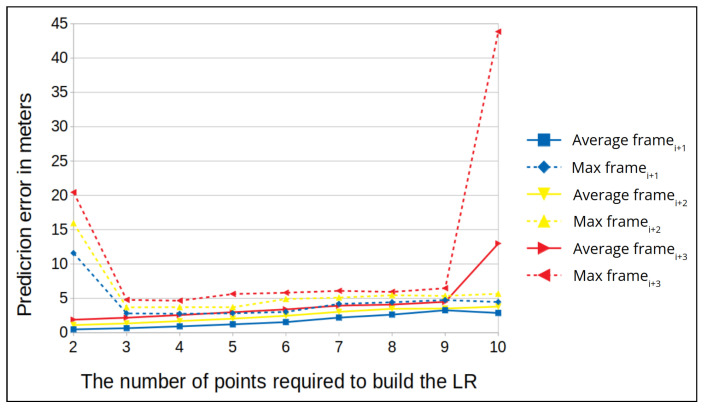
The prediction error (in meters) vs. various window sizes for an altitude of 50 m (average for the two locations).

**Figure 13 sensors-22-02608-f013:**
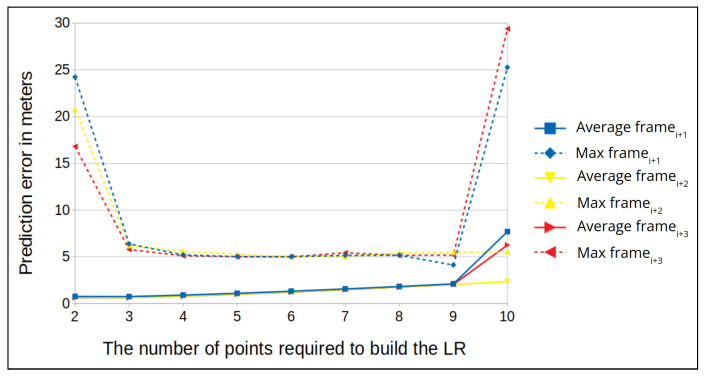
The prediction error (in meters) vs. various window sizes for an altitude of 100 m (average for the two locations).

**Figure 14 sensors-22-02608-f014:**
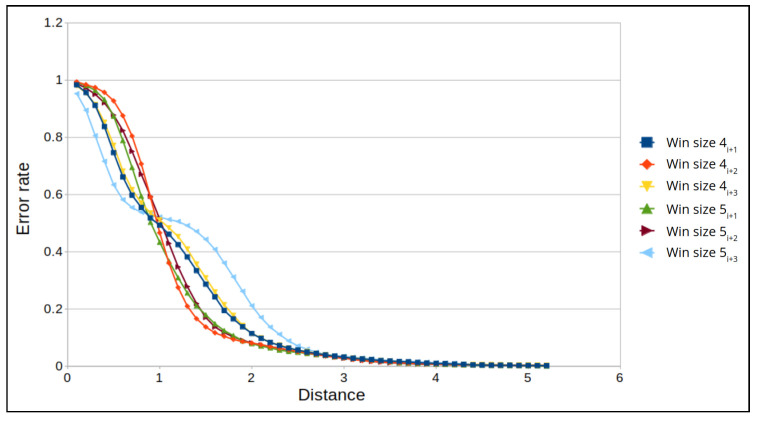
The FPR for a window size of 4 and 5 frames and an altitude of 50 m.

**Figure 15 sensors-22-02608-f015:**
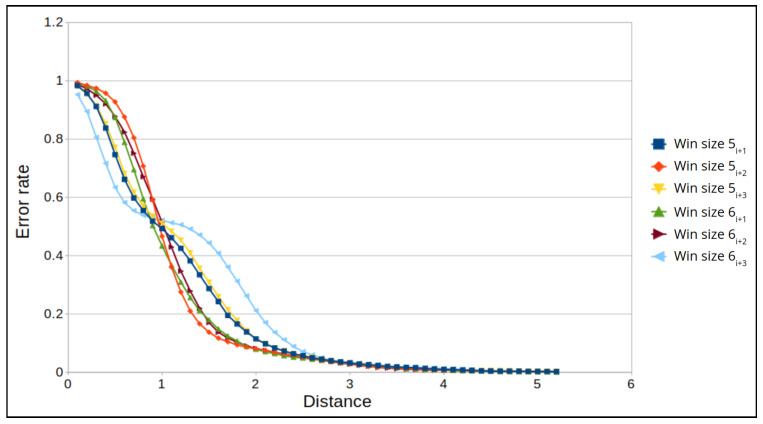
The FPR for a window size of 5 and 6 frames and an altitude of 100 m.

**Figure 16 sensors-22-02608-f016:**
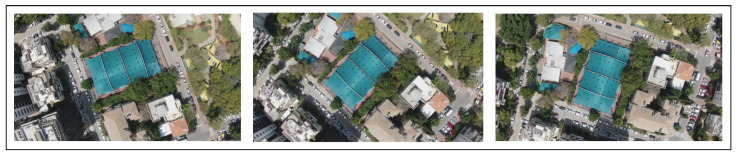
Example of a drone flight in which the drone is changing direction (rotating, rather than flying straight ahead), which causes a false positive due to the decreased correlation between the frames (the frames are presented from **left** to **right**).

**Figure 17 sensors-22-02608-f017:**
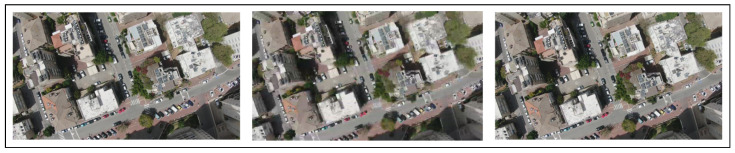
Example of three frames from a flight in a unchanging direction; the frame in the center is blurry; this causes a false positive due to the decreased correlation between the frames.

**Table 1 sensors-22-02608-t001:** The effect of a drone’s altitude on the performance; the results are presented for location #1 and various altitudes, with a drone speed of 5 m/s (18 km/h).

Altitude	Linear Function	RMSE	$R^{2}$	MEA
200 m	−0.224X + 97.984	1.2588	0.9984	0.9323
100 m	−0.464X + 93.165	3.8040	0.9952	1.4468
50 m	−1.072X + 100.797	5.7254	0.9934	2.0112

**Table 2 sensors-22-02608-t002:** The effect of a drone’s altitude on the performance; the results are presented for location #2 and various altitudes, with a drone speed of 5 m/s (18 km/h).

Altitude	Linear Function	RMSE	$R^{2}$	MEA
200 m	−0.220X + 97.168	1.9597	0.9974	1.2334
100 m	−0.497X + 103.025	3.7830	0.9958	1.6309
50 m	−1.229X + 107.230	39.3119	0.9664	5.6436

**Table 3 sensors-22-02608-t003:** The effect of a drone’s altitude on the performance; the results are presented for location #3 and various altitudes, with a drone speed of 5 m/s (18 km/h).

Altitude	Linear Function	RMSE	$R^{2}$	MEA
200 m	−0.221X + 95.962	1.8016	0.9976	1.1909
100 m	−0.500X + 101.049	2.1226	0.9976	1.1129
50 m	−1.125X + 99.657	11.3997	0.9881	2.9085

**Table 4 sensors-22-02608-t004:** The effect of a drone’s speed on the performance; the results are presented for location #1 and an altitude of 50 m, at various speeds.

Speed	Linear Function	RMSE	$R^{2}$	MEA
5 m/s (18 km/h)	−1.487X + 145.204	0.2551	0.9918	0.4591
10 m/s (36 km/h)	−0.878X + 91.951	1.5243	0.9878	1.1585
20 m/s (72 km/h)	−0.914X + 93.379	1.7241	0.9965	1.1379

**Table 5 sensors-22-02608-t005:** The effect of a drone’s speed on the performance; the results are presented for location #1 and an altitude of 100 m, at various speeds.

Speed	Linear Function	RMSE	$R^{2}$	MEA
5 m/s (18 km/h)	−1.900X + 187.233	5.8169	0.8138	2.3882
10 m/s (36 km/h)	−1.720X + 173.427	0.7974	0.9936	0.7550
20 m/s (72 km/h)	−1.818X + 179.150	2.9411	0.9941	1.6993

**Table 6 sensors-22-02608-t006:** The effect of a drone’s speed on the performance; the results are presented for location #1 and an altitude of 200 m, at various speeds.

Speed	Linear Function	RMSE	$R^{2}$	MEA
5 m/s (18 km/h)	−0.225X + 98.134	1.2764	0.9983	0.9385
10 m/s (36 km/h)	−0.230X + 99.831	0.2852	0.9996	0.4229
20 m/s (72 km/h)	−0.229X + 99.946	0.5753	0.9993	0.5700

**Table 7 sensors-22-02608-t007:** The effect of the flight terrain on the performance; the results are presented for location #2 and an altitude of 50 m, at various speeds.

Speed	Linear Function	RMSE	$R^{2}$	MEA
5 m/s (18 km/h)	−1.351X + 104.897	181.7630	0.8930	11.4522
10 m/s (36 km/h)	−1.105X + 97.275	247.5627	0.8547	13.2745
20 m/s (72 km/h)	−1.178X + 100.595	237.1031	0.8723	12.2619

**Table 8 sensors-22-02608-t008:** The effect of the flight terrain on the performance; the results are presented for location #2 and an altitude of 100 m, at various speeds.

Speed	Linear Function	RMSE	$R^{2}$	MEA
5 m/s (18 km/h)	−0.714X + 116.341	240.2957	0.8761	13.4783
10 m/s (36 km/h)	−0.726X + 117.098	250.6292	0.8748	14.0265
20 m/s (72 km/h)	−0.729X + 118.303	239.9461	0.8797	13.8666

**Table 9 sensors-22-02608-t009:** The effect of the flight terrain on the performance; the results are presented for location #2 and an altitude of 200 m, at various speeds.

Speed	Linear Function	RMSE	$R^{2}$	MEA
5 m/s (18 km/h)	−0.294X + 118.573	300.9058	0.8159	15.1959
10 m/s (36 km/h)	−0.279X + 115.805	278.9850	0.8254	14.5698
20 m/s (72 km/h)	−0.266X + 113.072	290.8093	0.8237	14.8485

**Table 10 sensors-22-02608-t010:** The effect of ambient light on the performance; the results are presented for an urban location, from an altitude of 50 m, under various lighting conditions (ranging from 75 to 10% light).

Light	Linear Function	RMSE	$R^{2}$	MEA
75%	−1.063X + 99.575	3.0514	0.9967	1.4503
50%	−1.111X + 102.380	6.8292	0.9926	2.1720
25%	−1.112X + 101.834	7.1568	0.9923	1.9989
10%	−1.026X + 125.131	543.1094	0.5925	19.7922

**Table 11 sensors-22-02608-t011:** The effect of ambient light on the performance; the results are presented for an urban location, from an altitude of 100 m, under various lighting conditions (ranging from 75 to 10% light).

Light	Linear Function	RMSE	$R^{2}$	MEA
75%	−0.485X + 96.490	3.4121	0.9956	1.4333
50%	−0.486X + 95.730	4.5149	0.9943	1.6018
25%	−0.500X + 99.148	4.7360	0.9943	1.8010
10%	−0.475X + 123.390	246.8611	0.7533	13.2136

**Table 12 sensors-22-02608-t012:** The effect of ambient light on the performance; the results are presented for an urban location, from an altitude of 200 m, under various lighting conditions (ranging from 75 to 10% light).

Light	Linear Function	RMSE	$R^{2}$	MEA
75%	−0.226X + 98.867	1.3956	0.9981	0.9657
50%	−0.231X + 99.090	2.4037	0.9970	1.2434
25%	−0.235X + 100.492	2.7024	0.9967	1.2696
10%	−0.225X + 103.089	41.5135	0.9487	5.8316

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
