# Peer review of "Towards the Detection of GPS Spoofing Attacks against Drones by Analyzing Camera’s Video Stream"

_sensors, 2022, doi:10.3390/s22072608_

Round 1

Reviewer 1 Report

VISAS - Detecting GPS Spoofing Attacks Against Drones by
Analyzing Camera’s Video Stream

[general]

this study presents some intuitive results that are confirmed by some controlled environment results
however, important practical aspects have not been considered or mentioned such
as fog, haze, water droplets, snow, lens fog,  and sudden strong gusts of wind (which understandbly, typically do not exist in the lab)

using thermal imaging could be an interesting avenue, that still requires exploration
in its own right (water bodies remain featureless/terrain) such as dynamic callibration perhaps
using the dji MSX functionality. 

the real world test at 4km/h unfortunately falls in a completely different bracket of comparison to the simulated studies (18km/h, 36km/h, 72km/h)
what happened? wind? if so, its ok to state that,
the 4km/h also raises issues of battery autonomy (~30min) 1km max range (for round trip) which can still be viable but could be mentioned.

i would recommend adding the word "Towards Detecting" TVISAS? in the beginning of the title

as fog, clouds, haze, snow, water droplets are big topics for a vision based system, not to mention how the linear regression will cope with
sudden gusts of wind without velocity vectors (also available in the drone)

please address/contextualize some of the concerns below.

[moderate]

line 147, please describe+cite the "similarity correlation"

There are several similarity measures, some even guaranteeing rotation invariance.
Dealing with fog and snow in the image similarity context is a hot topic
sometimes the fog or clouds are enough to make indestinguishable identical source images (one with cloud obstruction) from other completely
different source image in the similarity score (including using this instance of deep AI:
 https://deepai.org/machine-learning-model/image-similarity)

please describe how the current correlation is computed pixel by pixel
adding also a higher level description:
e.g.
if images are identical a value of 1 is given, is the images have partial overlap...

line 265 "Python technique?"

please include pseudo code

Table 4

please add throughout in brackets speeds in km, 5m/s (18km/h); 10 m/s (36km/h), 20m/s (72km/h)
so that it is "easier" to compare with the 4km/h of the results, which by the way
appears to be in a completely different bracket of comparison.

do attempt to contextualize this

please describe and characterize the origin of false positives that were found 

[minor]

line 36+118
MDPI citing convention
Sok, reference [1] in the West is the last author, I understand that perhaps it is the first author in other conventions?
I am 100% ok with that, I am just not sure what is the MDPI convention in such situations, or if there is a genuine mistake.

line 83
do you mean deaccelerate?

figure captions 4, 5, 6 left is 50 meters, right is 200 meters?

line 228

the wording can be improved (comparison of comparisons is currently stated just as a comparison lacking context),

suggestion: "At higher altitude the error at high speeds drops."

Reviewer 2 Report

Authors proposed a spoofing detection algorithm for drones based on cross-correlation between frames (from the drone's camera) and received GPS coordinates. The idea is sound and I really liked it: the more the drone moves away, the more the picture taken from the camera should be not correlated to the original position. The simulation/emulation methodology exploiting Google-Earth is interesting as well and provides a decent test for a real world scenario.

I have only one major concern about the paper. Authors mentioned many times to an algorithm to compute the correlation between images so called "Similarity correlation" but no formal definition is provided. This is the core of the idea. I would be very happy to see one or more images and the associated similarity as well as the "formal" way to compute it.

One last thing, authors missed to cite a few important works in the area that might be applied to the drone scenario as well.

[1] Gabriele Oligeri, Savio Sciancalepore, and Roberto Di Pietro. 2020. GNSS spoofing detection via opportunistic IRIDIUM signals. In Proceedings of the 13th ACM Conference on Security and Privacy in Wireless and Mobile Networks (WiSec '20). Association for Computing Machinery, New York, NY, USA, 42–52. DOI:https://doi.org/10.1145/3395351.3399350

[2] Gabriele Oligeri, Savio Sciancalepore, Omar Adel Ibrahim, and Roberto Di Pietro. 2019. Drive me not: GPS spoofing detection via cellular network: (architectures, models, and experiments). In Proceedings of the 12th Conference on Security and Privacy in Wireless and Mobile Networks (WiSec '19). Association for Computing Machinery, New York, NY, USA, 12–22. DOI:https://doi.org/10.1145/3317549.3319719

Reviewer 3 Report

Excellent and detailed analysis of existing solutions. Proper preparation for the experiment. The most significant limitation is the amount of light in the room. Despite all the limitations, this solution is appropriate and practicable.

The authors used the simulator to run experiments to determine the impact of various parameters on the method's performance. The root-mean-square error, R2, and mean absolute error were employed to evaluate the performance. It is now not entirely clear whether the method proposed by the authors is somewhat superior or simply a summary of the results.
